# An Intersectional Approach to Hepatitis B

**DOI:** 10.3390/ijerph20064879

**Published:** 2023-03-10

**Authors:** Christopher Lemoh, Yinzong Xiao, Lien Tran, Nafisa Yussf, Piergiorgio Moro, Sophie Dutertre, Jack Wallace

**Affiliations:** 1Department of Medicine, Melbourne Medical School, The University of Melbourne, Parkville, VIC 3010, Australia; 2Burnet Institute, Melbourne, VIC 3004, Australia; 3WHO Collaborating Centre for Viral Hepatitis, Peter Doherty Institute for Infection and Immunity, Parkville, VIC 3010, Australia; 4Department of Infectious Diseases, The University of Melbourne, Melbourne, VIC 3004, Australia; 5Centre for Culture, Ethnicity and Health, North Richmond Community Health, Richmond, VIC 3121, Australia; 6Centre for Social Research in Health, University of New South Wales, Kensington, NSW 2052, Australia

**Keywords:** hepatitis B, intersectionality, migrant health, refugee health, settler-colonialism, social epidemiology, social determinants of health, fundamental causes of disease, public health

## Abstract

Hepatitis B is a chronic condition, primarily associated with hepatitis B viral infection in early life. The failure of prevention and appropriate management can lead to subsequent liver cirrhosis and cancer. Hepatitis B most commonly affects people born in Asia and Sub-Saharan Africa and their global diasporas. The physical, psychological, and social impacts of hepatitis B are strongly influenced by sex and gender. Inequities in access to timely, sensitive diagnosis and effective management arise from interactions between structural inequalities related to race, ethnicity, Indigenous/settler status, class, and geography. The biomedical response to hepatitis B has led to advances in prevention, diagnosis, and treatment, but many affected communities have explanatory health belief models that differ from that of biomedicine. We argue that an intersectional approach, led by affected people and communities, can integrate biomedicine with the lived experience and social context that give purpose to and shape all personal, communal, clinical, and public health responses to hepatitis B. This approach has the potential to enable a consciously equitable, effective response to the biopsychosocial complexities of hepatitis B, improve the health and wellbeing of people living with hepatitis B, and reduce hepatitis B-associated mortality.

## 1. Introduction

Hepatitis B is a chronic infection presenting challenges of biological complexity and social inequity. Meeting these challenges requires the efforts of affected individuals and transnational communities, scientists, clinicians, and policymakers. We argue that an intersectional approach can provide a conceptual framework within which to create shared understandings between stakeholders with different models of health and illness, approaching from diverse perspectives and simultaneously dealing with other complex health issues.

## 2. Intersectionality

Intersectionality is a knowledge project viewing various systems of social difference as interacting and reinforcing each other to create social inequalities. It offers a method of thinking about the structural forces affecting our ways of living and moving through the world, paying particular attention to power gradients and differences in social status [1,2]. These forces influence vulnerability to infection and engagement with health services; they shape societal discourses concerning public health problems.

As a paradigm, intersectionality gives us a framework within which to think about problems: it sets conditions determining what is possible for us to imagine [3,4]. An intersectional approach centres on both the experience and praxis of individuals and groups who are negotiating multiple interacting inequities, supporting them to analyse and prioritise their needs, strategically leverage identities, and lead the creation of solutions to address their most pressing problems. They can form coalitions focusing on issues of mutual concern and mobilise for action, using whatever complementary social and physical resources are accessible through their various networks to push back against oppressive forces [1,5].

## 3. Intersectionality and the Social Determinants of Health

There is a fundamental causal relationship between social disadvantage and illness. Socioeconomic disparities cause multiple disease outcomes, through unequal access to resources promoting health or providing health care, frustrating disease-specific interventions that fail to address underlying inequities or social determinants of health [6,7,8]. Current public health thinking puts rectifying social disparities at the heart of improving public health, as did one of the early leaders in social epidemiology: Rudolf Virchow [9,10,11]. More recently, the Australian Institute of Health and Welfare (AIHW) has reported a clear association between disadvantaged social environments and chronic conditions such as obesity, chronic obstructive pulmonary disease, diabetes, and cancer [12]. This is clearly relevant to the chronic condition that is hepatitis B.

The social determinants of health framework sets out the broad structural inequities perpetuating global health problems but is still limited by its ostensibly objective perspective. An intersectional approach transcends this framework, going beyond simple description and advocacy to robustly address structural power relations with a critical view of truth-making and an ethical commitment to social justice, all the while maintaining a reflexive self-critical attitude, aware of its own implication in the power structures it critiques [4].

This approach is in fact consistent with one of Australia’s most successful public health innovations—the partnership response to HIV. This response took shape before the term “intersectionality” was coined, but viewing the partnership through the prism of an intersectional perspective has the potential to apply the hard-earned lessons from the HIV pandemic to the equally complex but importantly different pandemic of hepatitis B. Adopting an intersectional perspective for understanding hepatitis B as a physical and psychosocial issue enables the framing of a more diverse and inclusive array of questions answerable through systematic enquiry including (but not limited to) science and biomedicine.

## 4. Hepatitis B

Hepatitis B affects an estimated 296 million people worldwide (3.5% of the global population), with attributable deaths ranging from 555,000 to 887,000 annually [13,14]. Hepatitis B most commonly affects populations in Asia and Sub-Saharan Africa, many of whom are served by postcolonial health systems with multiple historical and current economic, social, and political challenges, amongst which hepatitis B is only one [15,16,17].

Chronic hepatitis B has a significant impact on the health of the Australian population, with almost 222,600 people living with hepatitis B in 2020 and 364 attributable deaths [18]. The infection is largely asymptomatic but can progress to chronic liver disease, cirrhosis, and hepatocellular carcinoma. Appropriate management prevents hepatitis-related deaths, but although safe and effective treatments are available in Australia, there is currently no cure [19]. The burden of chronic liver disease is predicted to rise, with the arrest of recent declines in hepatitis B-related deaths, unless diagnosis and treatment uptake substantially increase from current levels [20,21].

Clinical guidelines such as the Gastroenterological Society of Australia (GESA) 2022 Consensus Statement describe population groups with a high prevalence of chronic hepatitis B: essentially all non-Indigenous settlers should be screened for hepatitis B infection except those born in Australia, New Zealand, the Americas, Northwest Europe, or South Asia; screening for hepatitis B is recommended for Indigenous settlers, e.g., Māori migrants to Australia [22]. Screening recommendations from the American Association for the Study of Liver Diseases (AASLD) are even broader [23].

## 5. Complexity of Hepatitis B

### 5.1. Biomedical Complexity

The biomedical model remains the dominant influence in discourse about hepatitis B [22,24]. The problem and its impact are conceptualised and discussed in physical or biomedical terms: a viral particle infects an organ—the liver—of a human body; its genetic material integrates itself with human hepatocyte genes; viral enzymes lead to the production of various viral components, some of which are detectable by blood tests and some of which are themselves infectious viral particles that can enter the bodies of other humans through exposure to blood (most often during childbirth) or genital secretions. The presence of viral genes and proteins provokes an immunological response that can cause chronic liver inflammation and eventually fibrosis and cirrhosis. The presence of viral genetic material in human hepatocytes can disrupt cellular replication and lead to liver cancer.

This biomedical model is a powerful and useful way of conceptualising hepatitis B. It has led to effective vaccines; tests for diagnosis and monitoring disease activity and organ damage; tools to predict the risk of liver damage or liver cancer; and medications to interrupt viral replication or trigger an immune response that brings chronic infection to an end. However, hepatitis B is not merely a physical problem with a chemical solution, occurring within an individual body, without the cultural, familial, economic, or political context.

### 5.2. Psychosocial Complexity

Most people living with or affected by hepatitis B are members of dynamic, multigenerational, transnational social networks, but public health and clinical responses to hepatitis B in Australia focus largely on individuals and populations living within Australia. This downplays the lived experience of hepatitis B as a familial, intergenerational, chronic, and sometimes fatal condition that in many communities and countries can result in significant social marginalisation and limit economic and educational opportunities [25,26,27].

This is particularly important for cultural groups with well-established models of health where “the liver” may be conceptualised differently to Western biomedical science and medicine. Bryant et al. note the challenge of articulating biomedical explanations of hepatitis B as a “virus that invades bodies” for people whose primary language does not have a word for “hepatitis” or “virus”—particularly when providing information about the infection. This is further complicated by the lived experience of hepatitis B, understood and individualized as a cluster of symptoms (such as jaundice) or as a consequence of generational misfortune or breaching social norms [28].

Health systems are expressions of culture: professional biomedical systems inevitably engage with what Kleinman called popular and folk health systems [29]. It is therefore unsurprising that many of the communities most affected by hepatitis B use explanatory health belief models differing from the biomedical. Explanations of hepatitis B-related liver disease can include war, poor living standards in early life, a curse, immoral behaviour, diet, or shock [27,30]. The difference from a biomedical understanding is extended by the absence of physical symptoms for most people with hepatitis B for whom illness is only a concern where symptoms exist.

There is more to hepatitis B infection than its physical dimension: like other illnesses, it also has a psychological and social reality, including a moral dimension [29,31]. Subjective and shared meaning constitute lived experience: diagnosis; managing the condition; its impact on interpersonal relationships and interactions with structural forces shaping social reality. Apart from their intrinsic importance as determinants of suffering and other affective aspects of living with hepatitis B, these psychosocial dimensions affect the access of people to opportunities for preventing exposure and transmission and to the health care services, diagnostic tests, and medication that enable a person living with hepatitis B to remain well and avoid distressing or life-threatening complications of the condition. Hepatitis B—or rather, the awareness of its diagnosis—may change a person’s sense of self and their ability to live a fulfilling life as a member of their society. The psychosocial impact of the diagnosis may be felt before any physical symptoms through the imposition of formal restrictions on migration, employment, and recreation, or through informal restrictions such as acceptability as an intimate partner or spouse [26,27]. Another burden following diagnosis is the time commitment, economic burden, and psychological stress of prolonged engagement with the health system, often in the absence of symptoms or pharmaceutical treatment [32].

## 6. Intersectionality and Hepatitis B

An intersectional approach to a complex health problem in Australia such as hepatitis B needs to consider gender, class and geography, Indigenous/settler status, and nationality/residency status. These are all axes along which Australian society is inequitably divided in terms of access to social and material resources related to health and wellbeing, including access to timely, sensitive diagnosis, and long-term clinical management of hepatitis [33,34].

### 6.1. Gender

The physical, psychological and social impacts of hepatitis B are strongly influenced by sex and gender: men have higher rates of infection, progression to chronic infection, and complications, related to biological and social factors [35]. However, women are frequently diagnosed through antenatal screening, which provides access to treatment but presents psychosocial challenges such as health risks to unborn children and exposure to stigma and discrimination due to presumed immorality [25,27,35].

### 6.2. Class and Geography

Class must be considered when addressing social inequity. The myth of Australia’s “classless society” is no longer tenable, given that Australia now has several generations of inherited privilege and disadvantage that affect the ability of Australian residents to remain healthy or to access care when they fall ill [36]. In this respect, hepatitis B resembles other chronic health issues, which are very unevenly distributed across Australian society [37]. Both the prevalence of hepatitis B and the access/uptake of care are unevenly distributed across Australia, suggesting that the social environment is relevant, although information is not systematically collected or reported. The demographic characteristics of local populations, health infrastructure, and geography may play a part but routinely reported jurisdictional data are not linked with individual treatment outcomes, limiting the understanding of causal relationships or opportunities to intervene [33].

### 6.3. Settler-Colonialism

Hepatitis B disproportionately impacts people of colour, migrants, and Indigenous communities [14,18,22,38]. A bilingual survey of immigrants to Australia from mainland China found knowledge gaps regarding hepatitis B had not improved with the duration of residence in Australia, suggesting existing health promotion efforts had a limited impact [39]. An earlier survey of ethnically diverse people with hepatitis B found knowledge of risk factors, transmission, and treatment was associated with English proficiency, despite the widely acknowledged need for health promotion materials focusing on culturally and linguistically diverse (CALD) communities [40].

Like other anglophone nation-states (Canada, the United States, New Zealand) Australia originated as a settler-colonial project of the British Empire. Colonisation involved the invasion and occupation of territory as its prime objective, with the elimination of the native as a core organising principle [41]. This means the social institutions of Australia—including the health system, medical profession, and their underpinning scientific knowledge—developed within this imperial settler-colonial project. The language, symbolism and conceptual frameworks of medicine, public health, and epidemiology cannot be viewed as independent from it. There is thus a tension between settler-colonial science or health care and Indigenous ways of knowing and being [42].

### 6.4. Structural Racism and Cultural/Linguistic Diversity

There is also a tension between the settler-colonial state and the culturally diverse subjects constituting both the majority of people living with hepatitis B and an increasingly prominent part of the health workforce. The UK is still the most common country of birth for Australian migrant settlers, but it is rapidly being approached by India, China, and the Philippines; communities from regions of high hepatitis B prevalence are amongst those most commonly speaking languages other than English at home (Mandarin, Arabic) and with low English proficiency (Khmer, Vietnamese) [43]. Australia’s history of explicitly racist migration policies, intended to maintain the hegemony of white British settlers, radically changed during the mid-to-late-twentieth century, with consequently increased cultural diversity. Inequalities persist in which country of origin, ethnic identity, language, and class seem to play their parts, but the nature and understanding of “Anglo” social hegemony are not clearly articulated [44].

Australia’s medical workforce is increasingly diverse, with international medical graduates disproportionately practising in regional areas [45]. The nursing workforce is similarly diverse, with India, China, and the Philippines remaining prominent sources of recruitment and settlement [46].

Patients, clients, and their families accessing health care are increasingly culturally and linguistically diverse, presenting challenges in clinical communication as they receive care for health conditions arising from social inequities [47,48]. Some also experience systemic discrimination based on their nationality or residency status, such as eligibility for subsidised health care consultations and diagnostic tests through Medicare or access to affordable medications through the Pharmaceutical Benefits Scheme [49]. However, the collection and reporting of demographic data related to migration and cultural diversity are incomplete and inconsistent [50]. Non-intersectional analyses of these incomplete datasets may mask health inequities related to systematic discrimination using proxy markers of vulnerability such as country of birth [51].

The awkward term “culturally and linguistically diverse” (CALD) is used to define several groups of settlers labelled as key populations–but the term “CALD” is ambiguous and may not correlate or resonate with subjective group identity [52]. A recent paper from the AIHW has discussed how aggregated health data using demographic variables marking cultural and linguistic diversity can mask important differences in subjective wellbeing, burden of illness, and mortality rates. Australian-born, English-speaking people had a higher burden of illness, worse self-reported health, and higher mortality rates; however, the worst self-reported health and highest mortality rates were amongst people speaking Eastern European languages at home, hinting at the long-term consequences of inequities related to migration to a country where English is the dominant language, during the era of the White Australia policy; this stands as a warning for what may lie in store for other CALD migrants with chronic illness, once the “healthy migrant effect” has played out [53].

Interventions relying on external categorisations such as ethnicity have been piloted to improve screening for hepatitis B in populations with higher risk but have had little success [54]. Meanwhile, other guidelines and well-intentioned calls for inclusivity and the rectification of inequities affecting racialised populations such as Sub-Saharan African diasporas in high-income countries still uncritically use terms such as “Caucasian”, apparently operating within a discredited biological paradigm of race [55,56].

### 6.5. Transnational Diasporas

The concept of “diaspora” is a useful way to align population-level observations and interventions with the lived experience of individual subjects in their social networks. Diasporas form through displacement into occupied territory, in which Indigenous peoples have been displaced, suppressed, and repressed. Diasporas are composed of cultural formations which cut across and interrupt the settled contours of race, ethnos, and nation, bearing traces of their particular histories, cultures, and worlds of meaning [57,58]. In the case of hepatitis B, the transnational scope of the diaspora paradigm can encompass settings and events over the life course of individuals and families who experience perinatal infection in countries without widespread immunisation or antenatal screening, encounter systematic serology testing during migrant health assessments, and must negotiate education and employment conditions imposed through policy discriminating by hepatitis B serostatus [56]. It also frames the communication occurring within extended family and other social networks in Australia and countries of origin (and other countries of settlement) and exposure to media and health promotion material created in Australia and elsewhere, which shapes understandings of hepatitis B in the minds of those living with the condition and the people in their transnational social networks.

The Australian health system is heavily reliant on diasporic health professionals in the fields of nursing and medicine, as well as in support services such as catering, cleaning, and security. Added to the already gendered and hierarchical nature of health professions and the uneven power relations between the various professions, the element of diasporic cultural experience further increases the complexity of professional identity formation through educational discourses in healthcare settings [46,59].

## 7. Lessons from Australian Responses to HIV and Hepatitis C

### 7.1. Australian HIV Partnership

The Australian response to HIV has incorporated the awareness of the social determinants of health and illness, from the activism of affected communities to their partnership with health professionals, researchers, and policymakers [60,61,62,63]. This partnership enabled Australia to rapidly and flexibly develop effective health promotion to prevent infection and improve the uptake of testing, fast-track medications, and advocate for the inclusion of marginalised people into systems for sustainable, accessible treatment [64]. The close relationship between affected communities and specialist health professionals has seen rapid and gratifying advances in HIV and hepatitis C treatment effectiveness, availability, and uptake [65]. These achievements have involved intersectional identity work as described by Atewologun et al., as people with lived experience of HIV and hepatitis C have been heavily involved with key research initiatives and policy input, as well as health promotion and service provision [5]. However, progress on hepatitis B has been less impressive. There are important lessons from HIV and hepatitis C that can be applied to hepatitis B, but the differences are also significant.

### 7.2. Meaningful Engagement of Affected Communities

Two key elements driving success in HIV and hepatitis C are the commitment to meaningful engagement of affected communities and the centring of lived experience: peer-led organisations must actively engage with researchers and policymakers whilst maintaining credibility with their communities [65]. People with lived experience of HIV and hepatitis C sit on ministerial advisory bodies. Organisations representing affected individuals and communities receive government funding, make formal contributions to national strategies, provide professional development training to healthcare workers, and actively participate in scientific meetings such as the Australasian HIV/AIDS and Sexual Health Conference [66].

Populations affected by hepatitis B are larger, more heterogenous, more diverse, and geographically dispersed compared to those affected by HIV and hepatitis C (with which they overlap). Importantly, social networks of people affected by hepatitis B extend widely through Australian society, rather than being strongly identified with a self-aware, politicised community such as gay and bisexual men who, along with other LGBTQI groups, have driven much of the response to HIV since the early 1980s [61]. The group most heavily impacted in Australia by hepatitis C (people who inject drugs) remains heavily stigmatised and marginalised but is linked with health and community activists strongly advocating for treatment and support based equally on scientific evidence and human rights—many of whom were active in the HIV partnership.

### 7.3. Centring Lived Experience

The principle of meaningful engagement of people with lived experience has been central to the HIV response [67]. Its realisation has required ongoing struggle and has been hampered by a lack of conceptual tools to make visible and value the material contributions and impacts of such engagement, which has consequently relied heavily on inadequately supported volunteers. This deficit is now being addressed in the Australian context: the W3 framework provides conceptual tools for the description and measurement of peer leadership contributions in programmatic responses to HIV and hepatitis C [65]. This theoretical research is itself an example of such peer leadership.

It is only recently that peer-led activism has gained ground in hepatitis B, with the establishment of Hepatitis B Voices Australia: an organisation led by people with lived experience of hepatitis B, many of whom are also actively engaged as professional researchers or in health promotion (https://www.hepbvoices.au, accessed on 26 January 2023) [68]. This new approach was strongly in evidence at the 2022 Australasian Viral Hepatitis Conference in which people with lived experience of hepatitis B were involved as members of the organising committee, panellists, keynote speakers, and delegates, as well as research participants [66]. The conference was notable for its integration of diverse perspectives, including basic science, epidemiology, public health, interdisciplinary clinical professional, lived experience, and community advocacy.

For those who had been involved for many years in hepatitis B work, there was a palpable difference in approach which holds the promise for greater and more impactful collaboration in future. This approach was consistent with an intersectional paradigm but was not consciously articulated in such terms. There is a risk, therefore, that it may lack long-term sustainability, as individual practitioners and community members move on with their lives and careers, or that it may lack scalability and generalisability, without a theoretical framework which enables the creation of knowledge accepted as valid and authoritative in the academy, clinic, and community, permitting the mobilisation of social and material resources to further efforts to understand and respond to the challenges of hepatitis B.

## 8. Discussion

The biological, psychological, and social complexity of hepatitis B in relation to longstanding and fundamental social inequities suggests that an intersectional paradigm is well-suited to guide our response to this condition and its associated problems. An intersectional approach can guide inclusive collaboration for purposeful and strategic action, enable deeper awareness of diverse and relevant perspectives and create the capacity to deal with unforeseen problems and unintended consequences of public health and clinical interventions. The core principle is to centre people living with hepatitis B and affected communities within all policy, public health, research, health service provision, and other responses to the infection. People with lived experience must be meaningfully involved in decisions about setting priorities, allocating resources, planning action, and evaluating outcomes in terms that are relevant to those most affected. Tokenistic involvement and representation by “community leaders” will not suffice. We suggest that the commitment to centring lived experience be expressed in the following ways:Projects involving contact, communication, and collaboration with affected communities and people living with hepatitis B need working languages other than English. This means incorporating a budget for staff and technological support for multilingual consultation, planning meetings, documentation, and organisational activities. This should be underpinned by up-to-date population-level demographic data that includes publicly available information about relevant spoken and written languages, which can be used by local public health partnerships that include affected communities;There must be recognition that the experience of hepatitis B among culturally diverse communities differs within and across these communities given different traditional health belief models, social structures, health services access and migration and settlement patterns;More substantial health promotion is needed to raise community awareness and health literacy in a way to create political capital that directs adequate funding toward meaningful community-led stakeholder engagement in all aspects of hepatitis B research, health promotion, service planning/delivery and evaluation. There needs to be better visibility of this ongoing work in the public media space;Material support is needed to improve the accessibility of institutional activities for affected communities: increased subsidization for participation in conferences and educational/professional development meetings; live streaming of public meetings and conferences, with language services support;Hepatitis B must be included in discussions and activities undertaken in other sectors (education, housing, employment), to broaden engagement beyond health and maintain visibility of structural influences on hepatitis B epidemiology, as well as social impacts of public health responses.

A key stakeholder group in all this is the workforce of bicultural/bilingual community health workers who are employed by many organisations active in the hepatitis B space. These are members of affected communities, with relevant linguistic and cultural knowledge and communication skills across cultures and languages; they are well-placed to both mediate between communities, professionals, and institutions and take up leadership roles on projects and in institutions, if not prevented by inequitable power dynamics in the relevant organisations (both community-based and professional). Empowering and mobilising this workforce to provide more leadership may well be key to raising community awareness, mobilising political capital, and inducing organisations to improve their transparency, accountability, and cultural safety.

The recent development of professional guidelines and best practice standards for organisations working with bicultural workers may aid the empowerment of this workforce, to the benefit of all [69]. It will be important to address issues of remuneration and career progression as well as competence and professionalism, if prevalent inequities and exploitation are to be adequately addressed, enabling this workforce to fulfil its potential. This would require a peer-led community of practice and some form of representative organisation to advocate on behalf of its members to other stakeholders in the sector, including policymakers.

## 9. Conclusions

Hepatitis B is a complex social and public health challenge in Australia. It affects a diverse and heterogeneous range of communities and individuals, many of whom face multiple health and social challenges extended across transnational space and intergenerational expanses of time. The transmission, exposure, diagnosis, care, treatment, and lived experience of hepatitis B are shaped by dynamic interactions between systems of power relations affecting life chances and agency. Understanding the specific challenges facing people living with hepatitis B requires an intersectional approach that places people with lived experience at the centre to develop sustainable, community-led, meaningful stakeholder engagement. This demands that we ask the right questions, formulate useful answers, and create effective strategies to enable people to remain well and reduce the impact of hepatitis B on their physical and social health and wellbeing. The framework underpinning the response to this chronic infectious disease also has the potential to inform responses to hepatitis B beyond Australia, or address other complex, chronic health problems.

## Data Availability

Not applicable.

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
