# Peer review of "An Intersectional Approach to Hepatitis B"

_ijerph, 2023, doi:10.3390/ijerph20064879_

Round 1

Reviewer 1 Report

This is a timely and important article. The authors do an excellent job of leveraging appropriate public health theory and research on intersectionality and the ecological constructs associated with hepatitis B infection in Australia. This article will be meaningful not just for Australia, but globally. Please see a few of comments on the manuscript, which I hope will enhance the manuscript.

Author Response

We appreciate the positive comments made by the reviewer.

Please note the structure of the article has changed in response to the second reviewer’s comments, but the substance of the first reviewer’s comments have been addressed. 

Introduction:

  1. We have rephrased the statement about the usefulness of an intersectional approach to make it clear that this is the argument we make in this article.

Inequity of hepatitis B

  1. We have defined the term described the the “CALD” acronym - now in Section 6 (Intersectionality and hepatitis B)

Understanding the complexity of hepatitis B

  1. We have deleted the term “close contact” from the section on transmission  - Section 5 (Complexity of hepatitis B)
  2. We have added “immoral behaviour” to the ist of folk health beliefs  - Section 5 (Complexity of hepatitis B)
  3. We have specified “physical symptoms”  - Section 5 (Complexity of hepatitis B)
  4. We have defined “AIHW” - now in Section 3 (Intersectionality and the social determinants of health)
  5. We have attended to typographical errors

Section 7 (Lessons from Australian responses to HIV and hepatitis C)

  1. Regarding legislative commitment to addressing HIV and HCV epidemics, we would argue that such legislation was a consequence of meaningful engagement of affected communities. Given the length of the manuscript we did not go into detail in this regard.

Discussion

  1. We thank the reviewer for making the important point about the need for resourcing of bilingual/bicultural worker collective action - we have added a section to specifically mention the issue of remuneration.

Once again we thank the reviewer for their generous comments and helpful suggestions.

Reviewer 2 Report

Thank you for the opportunity to review. The authors make a thoughtful and novel contribution. This article attempts to draw together several ideas and frameworks, which at times made it difficult to read. I suggest some slight reorganising of the manuscript and further signposting might help clarify ideas.

The section on ‘Intersectionality and hepatitis B’ was, at times, difficult to read and may benefit from some further sub-headings. I think it would be useful to situate and briefly describe intersectionality earlier in the manuscript. This section, for me, was largely descriptive and could have benefited with getting to the crux of intersectionality: how do these social factors and structures of power intersect in relation to hepatitis B?  

The purpose of section 5 was lost on me. While the critique of SDOH was interesting, it wasn’t clear what the relation to hepatitis B specifically was. I am also unsure if this section is saying that the response to HIV utilised an SDOH approach, or an intersectionality approach, or both. A more explicit statement on how SDOH does/doesn’t support the response to Hep B would be helpful – or perhaps revaluate what this section adds to the manuscript.

The authors provide a good overview of the HIV response, and contribution of people with lived experience (could mention MIPA and lack of for Hep B). Again, this section could benefit with more explicit statement as to how meaningful involvement facilitates an intersectional approach (and/or vice versa).

I agree with the recommendations proposed by the authors. I wonder if it would be helpful to explicitly name who may be responsible for both advocating for, and implementing, these recommendations. Secondly, these recommendations focus on involvement of people with lived experience at a ‘practice’ level. While I think this is a good first step, I think the lesson from the HIV response is the importance of people with lived experience involved in decision-making. It would be useful to have a recommendation (or a statement) that proposes the importance of this (if the authors agree), and a vehicle by which this could be achieved.

Author Response

We thank the reviewer for their very helpful comments and suggestions.

We have restructured the article and introduced more subheadings to signpost themes in the argument.

  • We have led with a description of intersectionality (Section 2). We have followed this with a description of the social determinants of health (Section 3) and an explanation of how intersectionality builds upon and transcends the SDOH paradigm, through its self-reflexive critique of power, which is important for the mobilisation of marginalised social groups.
  • We argue that the Australian HIV partnership was in effect and intersectional response, although that terminology was not used at the time (Section 3). We argue that using intersectionality as a paradigm to reflect on experience of the HIV and hepatitis C response will enable us to apply the lessons learned from previous pandemics to the hepatitis B pandemic, whilst remaining sensitive to its important differences from HIV and HCV.
  • We have made clearer mention of the lack of meaningful involvement of people living with hepatitis B (Section 7), but have elected to retain the consistent terminology of “meaningful experience of affected communities” rather than the historical and more specific “meaningful involvement of people living with HIV/AIDS”.
  • We have kept the recommendations broad in order to maintain their relevance to the wide variety of stakeholders in the hepatitis B response and to focus on the principles of meaningful engagement of affected communities and centring lived experience (Section 8).
  • We have made specific mention of the need to centre lived experience in decision-making, not just practice (Section 8).
  • We have not suggested a vehicle for for decision-making, as this might best be led by groups of people with lived experience, such as the Hepatitis B Voices group mentioned in our article. We have, however, been more explicit in our call for bicultural workers to organise for action in both community advocacy and professional status (Section 8).

We believe that the changes to the article suggested by the reviewer have enhanced its quality and appreciate the time taken.